# Elucidating Sequence and Structural Determinants of Carbohydrate Esterases for Complete Deacetylation of Substituted Xylans

**DOI:** 10.3390/molecules27092655

**Published:** 2022-04-20

**Authors:** Leena Penttinen, Vera Kouhi, Régis Fauré, Tatiana Skarina, Peter Stogios, Emma Master, Edita Jurak

**Affiliations:** 1Department of Chemistry, University of Eastern Finland, 80130 Joensuu, Finland; leena.penttinen@uef.fi; 2Department of Pharmacy, University of Helsinki, 00790 Helsinki, Finland; vera.kouhi@helsinki.fi; 3Toulouse Biotechnology Institute, Bio & Chemical Engineering, Université de Toulouse, Centre National de la Recherche Scientifique, Institut National Des Sciences Appliquées, 31077 Toulouse, France; regis.faure@insa-toulouse.fr; 4Department of Chemical Engineering and Applied Chemistry, University of Toronto, Toronto, ON M5S 3E5, Canada; tanskarina@gmail.com (T.S.); p.stogios@utoronto.ca (P.S.); emma.master@utoronto.ca (E.M.); 5Department of Bioproducts and Biosystems, Aalto University, 02150 Espoo, Finland; 6Department of Bioproduct Engineering, University of Groningen, 9747 Groningen, The Netherlands

**Keywords:** acetyl xylan esterase, carbohydrate esterase, xylan, SGNH hydrolase

## Abstract

Acetylated glucuronoxylan is one of the most common types of hemicellulose in nature. The structure is formed by a β-(1→4)-linked D-xylopyranosyl (Xyl*p*) backbone that can be substituted with an acetyl group at *O*-2 and *O-*3 positions, and α-(1→2)-linked 4-*O*-methylglucopyranosyluronic acid (MeGlc*p*A). Acetyl xylan esterases (AcXE) that target mono- or doubly acetylated Xyl*p* are well characterized; however, the previously studied AcXE from *Flavobacterium johnsoniae* (*Fjo*AcXE) was the first to remove the acetyl group from 2-*O*-MeGlc*p*A-3-*O*-acetyl-substituted Xyl*p* units, yet structural characteristics of these enzymes remain unspecified. Here, six homologs of *Fjo*AcXE were produced and three crystal structures of the enzymes were solved. Two of them are complex structures, one with bound MeGlc*p*A and another with acetate. All homologs were confirmed to release acetate from 2-*O*-MeGlc*p*A-3-*O*-acetyl-substituted xylan, and the crystal structures point to key structural elements that might serve as defining features of this unclassified carbohydrate esterase family. Enzymes comprised two domains: N-terminal CBM domain and a C-terminal SGNH domain. In *Fjo*AcXE and all studied homologs, the sequence motif around the catalytic serine is Gly-Asn-Ser-Ile (GNSI), which differs from other SGNH hydrolases. Binding by the MeGlc*p*A-Xyl*p* ligand is directed by positively charged and highly conserved residues at the interface of the CBM and SGNH domains of the enzyme.

## 1. Introduction

Hemicelluloses represent the second most abundant polysaccharide in plant fiber (i.e., lignocellulose), after cellulose. The composition and structure of hemicelluloses vary depending on botanical source. 4-*O*-Methyl-glucuronoxylans (GXs) comprise approximately 20% of wood fiber from deciduous trees [1,2] and are characterized by a β-(1→4)-linked D-xylopyranosyl (Xyl*p*) backbone that can be acetylated and/or substituted by α-(1→2)-linked 4-*O*-methyl/D-glucopyranosyluronic acid (MeGlc*p*A) at a reported frequency of 1 MeGlc*p*A for every 10–15 Xyl*p*, and acetyl units at *C*-2 and/or *C*-3 positions [2,3,4]. The presence of such side groups can impact xylan properties, including water solubility, rheology, and nutrient value, and can hinder the digestibility of xylan and the access of xylan-degrading enzymes [5,6,7,8,9]. Accordingly, enzymes that selectively remove xylan side groups are important for tailoring xylan structures and increasing the accessibility of main-chain xylanases that convert xylan polymers to oligomers or monomeric xylose. Whereas tailored xylan polymers can be processed into marketable materials, such as packaging films and hydrogels [6,10,11,12], further enzymatic deconstruction of xylan to xylo-oligosaccharides and xylose is needed for the production of prebiotics, low caloric sweeteners such as xylitol, and fermentation products [13,14].

Enzymes targeting side groups of GX have been classified into multiple carbohydrate-active enzyme families (CAZymes) [15]. These include α-glucuronidases from glycoside hydrolase (GH) families GH67 and GH115, and acetyl xylan esterases (AcXEs) that have been primarily reported in carbohydrate esterase (CE) families CE1, CE4–CE6, and CE16 [15]. Whereas α-glucuronidases from family GH115 hydrolyze the α-(1→2)-linkage between MeGlc*p*A and mono-substituted Xyl*p* at internal and end positions of GX [16,17,18,19,20], GH67 activity is restricted to such linkages at the non-reducing end of xylans [21,22,23]. CEs acting on GX include those that act on *O*-2 and *O*-3 monoacetylated Xyl*p* (CE2, CE3, CE4), and others with preferential activity towards 2,3-di-*O*-acetylated Xyl*p* (CE1, CE5, CE6) [24]. Certain family CE16 enzymes were previously shown to act on non-reducing MeGlc*p*A-(1→2)-3-*O*-acetyl-Xyl*p* positions [24,25], possibly as a result of acetyl group migration from *O*-3 to *O*-4 on the non-reducing Xyl*p* [26]. In 2018, we identified an unclassified carbohydrate esterase from *Flavobacterium johnsoniae* (*Fjo*AcXE) with the ability to deacetylate MeGlc*p*A-(1→2)-3-*O*-acetyl-Xyl*p* subunits at internal positions of the xylan backbone [27]. This finding was subsequently reported for a second acetyl xylan esterase from *Flavobacterium johnsoniae* that shares over 98% identity with *Fjo*AcXE, but from a different *F. johnsoniae* strain [28]. Prior to its characterization, *Fjo*AcXE was annotated as a protein of unknown function (PUF) belonging to the polysaccharide utilization locus (PUL) PUL20 of *F. johnsoniae* UW 101, which comprises CAZymes predicted to act on xylans including sequences belonging to families GH43, GH115, CE1 and CE6 (http://www.cazy.org/PULDB/, accessed on 13 February 2022) [29]. The *Fjo*AcXE sequence was predicted to include a signal sequence for secretion and a SGNH hydrolase-type esterase domain belonging to GDSL-like lipase/acylhydrolase-type domain (Pfam domain PF12472). Structural models of *Fjo*AcXE showed an α/β/α-fold that is characteristic to SGNH hydrolases and an N-terminal CBM-like structure. Different to most similar esterases from families CE2 and CE12, three loops surrounding the predicted catalytic site of *Fjo*AcXE adopted unique conformations predicted to play a role in substrate preference and substrate binding. The found activity completes the deacetylation of xylan and was shown to boost α-glucuronidase activity by offering α-glucuronidase free access to xylan unit [27].

To expand our knowledge of MeGlc*p*A-(1→2)-3-*O*-acetyl-Xyl*p* acting CE enzymes and elucidate their structural characteristics, we selected six *Fjo*AcXE homologous that shared 47–98% sequence identity and produced them for structural and biochemical characterization.

## 2. Results and Discussion

### 2.1. Candidate Selection and Recombinant Protein Production

Homologous sequences of *Fjo*AcXE were retrieved through BLASTp search against GenBank and six sequences comprising Lipase_GDSL_2 (PF13472) and SGNH_hydro domains were selected for further studies (Appendix A). Alignment of the six selected sequences with *Fjo*AcXE (Figure 1) revealed that all homologs possess Ser, Gly, Asn, and His (SGNH) residues that are the signature feature of SGNH-type hydrolases, along with the conserved Ser, Asp, and His residues predicted to form the catalytic center. The six selected sequences were functionally expressed in *E. coli* and purified using affinity chromatography (Appendix A).

### 2.2. General Biochemical Properties

The recombinantly produced proteins showed no or negligible activity on all tested *p*NP-glycosides and selected commercially available polysaccharides (Appendix A). Instead, clear activity was observed for all targets on *p*NP-acetate (*p*NPA) and 4-MUA (Table 1). Given the pH stability of 4-MUA [30], this substrate was used to evaluate the pH optimum of each enzyme. In all cases, the pH optimum was between pH 6.0 and pH 7.5 (Appendix A), which is slightly lower than the pH optimum reported for *Fjo*AcXE (pH 8.0, 25). With the exception of *Aim*AcXE, all enzymes retained over 50% activity after 30 min at 55 °C (Appendix A) and thus displayed higher temperature stability than *Fjo*AcXE.

### 2.3. Activity on Acetylated Oligosaccharides

Based on the general biochemical properties of the targeted enzymes, activity studies using acetylated carbohydrates were performed at pH 7.0 and 30 °C. The per-acetylated xylooligosaccharides display acetyl substitutions at the reducing anomeric *O*-1 position, non-reducing *O*-4 position as well as all *O*-2 and *O*-3 positions of the Xyl*p* subunits. Given that the theoretical maximal yield of acetic acid was determined to be 3 μg/mL, approximately 60% of total acetyl groups were released by all selected enzymes after 20 h, including *Fjo*AcXE (Figure 2A). This apparent limit to the extent of deacetylation likely reflects the regioselectivity of the enzymes on the synthetic per-acetylated xylooligosaccharides. Moreover, these results confirmed that, like *Fjo*AcXE, the enzymes reported for the first time herein are carbohydrate esterases.

All selected enzymes also deacetylated xylooligosaccharides from corn fiber (CF-XOS) fraction (Figure 2B), albeit to a lesser extent than per-acetylated xylooligosaccharides. The theoretical maximal yield of acetic acid was calculated to be 1.2 μg/mL; therefore, less than 20% of total acetyl groups were released by all enzymes, including *Fjo*AcXE after 20 h. Previously, it was shown that *Fjo*AcXE is able to target 2-*O*-acetyl-Xyl*p* where the same Xyl*p* is substituted at the *O*-3 position with L-arabinosyl residues [27]. The CF-XOS used is expected to be prevalently feruloylated [31]. Given that *Fjo*AcXE was unable to release acetyl groups that neighbor feruloylated side groups, it can be speculated that the remaining acetyl groups belong to that CF-XOS.

### 2.4. Complementation of a-Glucuronidase Activity

AcXE activity on MeGlc*p*A-(1→2)-3-*O*-acetyl-Xyl*p* was evaluated by testing the synergistic effect of the AcXEs with GH115 α-glucuronidase from *Amphibacillus xylanus* (*Axy*Agu115A) on acetylated glucuronoxylan. Approximately ten-times higher MeGlc*p*A was released by α-glucuronidase after adding AcXE to the reaction (Figure 3). The result clearly shows that the studied AcXEs deacetylate MeGlc*p*A-(1→2)-3-*O*-acetyl-Xyl*p* of glucuronoxylan, and possibly to some extent 2-*O*-acetyl-Xyl from neighboring xylopyranosyl residues. This catalytic specificity is only found in the previously reported *Fjo*AcXE and the six homologs of this study. The apparent uniqueness of this activity motivated structural characterization of the studied CEs to uncover features that promote binding of the MeGlc*p*A-(1→2)-3-*O*-acetyl-Xyl*p* substrate.

### 2.5. Structural Determinants of the New CE Family

We attempted to crystallize *Fjo*AcXE and its homologs *Fsp*AcXE, *Fsp*F52AcXE, *Csp*AcXE, *Pbe*AcXE, *Fre*AcXE and *Aim*AcXE. We successfully solved the structure of the CBM-like domain of *Fjo*AcXE (to resolution 2.45 Å) and multiple structures of the full-length *Csp*AcXE and *Pbe*AcXE enzymes. Based on the protein sequence of *Fjo*AcXE, the CBM-like domains do not clearly group with the existing CBM families. The crystal structure of *Csp*AcXE was solved in the native state (1.30 Å). Three crystal structures of AcXE from *Prolixibacter bellariivorans* (*Pbe*AcXE) were also solved, including a native form (1.35 Å) and two complex structures, one with MeGlcpA-Xylp (to 1.26 Å) and another with acetate (1.13 Å). Full x-ray crystallographic statistics can be found in Appendix A.

The sequence identity between the *Csp*AcXE and *Pbe*AcXE is 52.4%. Both comprise an N-terminal CBM-like domain consisting of three antiparallel β-sheets, and a C-terminal SGNH domain where the central β-sheet comprises five parallel β-strands flanked on both sides with α-helices (Figure 4). In both cases, the CBM-like and catalytic SGNH domains are tightly packed and connected through a loop of six to seven amino acids. The residues forming the CBM-like domains in *Pbe*AcXE and *Csp*AcXE are 1–175 and 1–176, respectively, whereas residues forming the SGNH domain in *Pbe*AcXE and *Csp*AcXE are 180–385 and 181–386, respectively. The first seven N-terminal residues of *Pbe*AcXE and the first 10 residues of *Csp*AcXE did not have electron density and so were not included in the final structures. Moreover, residues 350 to 365 in *Csp*AcXE, which form a loop next to the active site, had unclear or missing electron density and so were omitted from the solved structure; this could be the result of the absence of ligand to stabilize its conformation, or due to the absence of many crystal packing contacts in this region of the protein.

As indicated by a structural similarity search against PDB using Dali (Figure 5), the structures of *Pbe*AcXE and *Csp*AcXE are most similar to structures of GDSL hydrolases of unknown specificity and some with esterase/lipase activity, i.e., TesA from *E. coli* and *Pseudomonas aeruginosa* [32] (Appendix A). There was also a clear structural similarity with known acetyl xylan esterases, including *Al*AXEase from *Arcticibacterium luteifluviistationis* (PDB 7ddy), Axe2 from *Geobacillus stearothermophilus* (PDB 4jko) [33], a CE3 enzyme from *Talaromyces cellulolyticus* (PDB 5b5s) [34], and CE2 enzyme from *Cellvibrio japonicus* (PDB 2waa) [35] and *Butyrivibrio proteoclasticus* B316 (PDB 3u37) [36] (Figure 5).

The CBM-like domains of *Fjo*AcXE, *Pbe*AcXE and *Csp*AcXE adopted a similar structure (Figure 4). The construct of *Fjo*AcXE that successfully crystallized did not include the first 36 amino acids that comprise the CBM; residues 37–40 and residues 137–148 were also disordered in the structure. Due to the incomplete *Fjo*AcXE CBM structure, further analysis will focus on the *Pbe*AcXE and *Csp*AcXE structures, although based on our analysis in subsequent sections, we expect that these regions would participate in the active site, i.e., Lys34 and Arg148 of *Fjo*AcXE. The CBM from these proteins was most similar to a CBM1 domain from GH10 enzyme from *Bacteroides intestinalis* (PDB 4qpw) [37], CBM35 from a GH31 enzyme from *Paenibacillus* sp. 598K (PDB 5x7s) [38], and the Ig-like domain from a GH86 enzyme from *Bacteroides uniformis* (PDB 5ta1) [39] and the β-sandwich domain from a GH87 enzyme from *Streptomyces thermodiastaticus* (PDB 7c7d) [40] (Dali Z-scores between 6.4 and 6.9, RMSDs 3.3–4.2 Å over 94–102 Cα atoms) (Appendix A). Based on a more detailed comparison with the structure of the CBM1 in complex with xylotriose (Appendix A), the CBM from *Pbe*AcXE and *Csp*AcXE does not appear to have an equivalent ligand binding pocket due to the positioning of the loops 73–82 and 146–160; however, conformational changes in this area in the presence of the appropriate ligand may open a binding pocket.

### 2.6. Sequence Determinants of the New CE Family

The active site of *Pbe*AcXE and *Csp*AcXE is located at the C-terminal end of the central β-sheet of the SGNH domain, as is common for all α/β/α hydrolases [41]. The Ser-Asp-His catalytic triad is formed by Ser188, Asp363, and His366 in *Pbe*AcXE, and Ser189, Asp363, and His366 in *Csp*AcXE. In both cases, the active sites are on the surface of the enzymes and highly exposed to solvent, where the serine occurs in a loop following the β1-strand, and the aspartate and histidine are positioned on a long loop between the β5-strand and a short α-helix (Figure 4). In *Pbe*AcXE, the oxyanion hole that stabilized the oxyanion intermediate during catalysis is formed by the main chain amides of Ser188 and Gly228, and the amine side group of Asn265 (Figure 6). In *Csp*AcXE, these residues are Ser189, Gly229 and Asn266.

The SGNH lipase/hydrolase-type proteins have four sequence blocks each containing a conserved catalytic or oxyanion hole residue: Serine in block I, glycine in block II, asparagine in block III and histidine in block V [42]. In SGNH hydrolases, especially lipases, the catalytic serine is often part of a GDSL sequence motif [42,43]. Interestingly, a GDSL motif is not found in AcXEs; however, a shorter GDS motif was found in an AcXE from *Geobacillus stearothermophilus* [44] and in an AcXE from *Caldicellulosiruptor bescii* [45]. In *Fjo*AcXE and all studied homologs, the sequence motif around the catalytic serine is Gly-Asn-Ser-Ile (GNSI). The most common consensus sequence in α/β hydrolases is GXSXG, which defines the nucleophilic elbow of an active site [46,47]. Therefore, like the GDSL-hydrolases, the GNSI-AcXEs do not have the nucleophilic elbow; this is corroborated by the crystal structures of *Pbe*AcXE and *Csp*AcXE, which show that the Asn in this motif (Asn187 and Asn188) are buried into the interior of the enzymes. Since the GNSI sequence motif has not been reported in any other carbohydrate esterase or lipases, this may be a signature feature of *Fjo*AcXE-like AcXEs.

### 2.7. Five Conserved Loops Shape the Active Site

The substrate-binding sites in *Pbe*AcXE and *Csp*AcXE are mainly shaped by five conserved loops, from which two belong to the CBM domain and three to the SGNH domain. The loops in *Pbe*AcXE are 16–35, 127–142, 185–203, 224–244, and 346–369; in *Csp*AcXE, the loops are 19–37, 127–144, 187–204, 227–242, and 349–368 (Figure 7). The 185–203/187–204 loop also contains one short α-helix. Here, the loops are named loop1 to loop5 by their order of occurrence in the sequence. All five loops include conserved regions in *Fjo*AcXE and the studied homologs. Loop1 is long and highly conserved but contributes only one residue (Asn26 of *Pbe*AcXE and Asn28 of *Csp*AcXE) to the active site. The GNSIXDGRGS region in loop3 includes the GNSI motif sequence stretches that overlap with block I of the SGNH blocks (i.e., residues 186–195 in *Pbe*AcXE and 187–196 in *Csp*AcXE). Loop4 overlaps with block II of the SGNH blocks and comprises two conserved stretches: GIGG and GGLGP (226–229 and 236–240 in *Pbe*AcXE, and 227–230 and 235–239 in *Csp*AcXE). Lastly, loop5 comprises the DXLHP motif, which include the catalytic aspartate and histidine in block V of the SGNH blocks (Asp363 in both *Pbe*AcXE and *Csp*AcXE; this region was unmodelled in *Csp*AcXE due to disorder).

### 2.8. Substrate Binding at the Active Site

Two complex structures of *Pbe*AcXE were solved, one with bound MeGlc*p*A-(1→2)-Xyl*p* and another with acetate. These two complex structures allowed for the examination of substrate binding in AcXEs that are active towards MeGlc*p*A-(1→2)-3-*O*-acetyl-Xyl*p*. The MeGlc*p*A-(1→2)-Xyl*p* ligand is bound into a positively charged and highly conserved (Appendix A) environment formed by Arg25, Asn26, Arg139 and Arg193 (Figure 6). Specifically, the carboxylic acid group of MeGlc*p*A is bound to Arg139 in the bidentate way and to the catalytic Ser188; the *O*-3 and *O*-5 oxygen of the Xyl*p* ring is also bound to Ser188.

In addition to coordination by Arg139, the hydroxyl groups at the *C*-2 and *C*-3 positions of MeGlc*p*A and *O*-5 of the pyranose ring are bound by Arg25, Asn26 and Arg193, respectively. At the same time, the Xyl*p* is further coordinated by Ser188, Arg193, and His366. Residues Arg25, Asn26 and Arg139 all belong to the CBM, whereas Ser188, Arg193 and His366 belong to the SGNH domain. This reveals that despite the catalytic site being localized to the SGNH domain, the CBM domain plays a role in binding MeGlc*p*A-(1→2)-3-*O*-acetyl-substituted xylan and enabling the unique activity of this type of AcXEs.

Interestingly, in the *Pbe*AcXE·MeGlc*p*A-(1→2)-Xyl*p* crystal we noticed weak additional density extending beyond the *O*-1 position of the Xyl*p* unit (Appendix A). We attempted to model two additional Xyl*p* units into this density but given the poor quality of the density they were not included into the final deposited structure.

The complex structure of *Pbe*AcXE with acetate shows that acetate is bound to the oxyanion hole by Ser188, His366, Asn265 and Gly229, and the α-carbon is localized in a hydrophobic pocket formed by Ile189, Val264 and Leu365. Based on a superposition of the *Pbe*AcXE·acetate complex with the *Pbe*AcXE·MeGlc*p*A-(1→2)-Xyl*p* complex, the location of the acetate corresponds to the O4 position of the bound Xyl*p*; this is consistent with *Pbe*AcXE’s recognition of the non-reducing end, i.e., MeGlc*p*A-(1→2)-4-*O*-acetyl-Xyl*p*. The catalytic Ser188 was modeled in two conformations, suggesting flexibility during catalysis. All other residues are conserved over the studied AcXEs except Arg25, which can be either replaced by histidine, lysine or asparagine.

## 3. Materials and Methods

### 3.1. Materials

All polysaccharides were purchased from Megazyme (Ireland) with the exception of oat spelt xylan, which was purchased from Sigma-Aldrich (St. Louis, MO, USA) (Appendix A). All para-nitrophenol (*p*NP) substrates and 4-methylumbelliferyl acetate (4-MUA) were purchased from Sigma (St. Louis, MO, USA) (Appendix A). The per-acetylated xylo-oligosaccharide mixture comprised chemically acetylated xylooligosaccharides (per-acetylated) with degree of polymerization between 4 and 7 D-xylosyl units [48]. Acetylated glucurono-xylooligosaccharides (Ac-XOS) were isolated by steam extraction of milled chips from Eucalyptus wood [49] and was a kind gift from Prof. Maija Tenkanen (University of Helsinki, Finland). Acetylated and feruloylated AcFaXOS from corn fiber (CF-XOS) was isolated by mild acid hydrolysis and was received as a kind gift from Prof. Mirjam Kabel (University of Wageningen, the Netherlands). The characterization of the CF-XOS (fraction B) was previously described by Appeldoorm et al. [31]. All other chemicals were analytical grade and were ordered from Sigma-Aldrich or Fisher Scientific. *Fjo*AcXE and *Axy*Agu115A were prepared as previously described [16,27].

### 3.2. Candidate Selection

Six homologues that share sequence identity between 47 and 98% with *Fjo*AcXE (accession number ABQ06890.1) were selected. The protein sequences are listed in Appendix A, and included selections from *Flavobacterium* sp. 40S8 (WP_073411463.1), *Flavobacterium* sp. F52 (WP_008467835.1), *Chryseobacterium* sp. YR480 (*Csp*AcXE, WP_047423182.1), *Prolixibacter bellariivorans* (*Pbe*AcXE, WP_027586161.1), *Flavobacterium reichenbachii* (*Fre*AcXE, WP_035683315.1), and *Alkaliflexus imshenetskii* DSM 15055 (*Aim*AcXE, WP_044117739.1).

### 3.3. Protein Expression and Purification

For biochemical studies, selected genes were synthesized and cloned into pET-29b + plasmid (C-terminal His-Tag) by Genscript (Piscataway, NJ, USA) and then transformed into *E. coli* BL21(DE3) for protein expression. Transformants were propagated at 30 °C in 2 L Erlenmeyer flasks containing 500 mL of Luria-Bertani (LB) Broth-Miller supplemented with 50 µg/mL kanamycin, 0.5 M sorbitol and 0.025 M glycine betaine. When cultures reached an OD600 of ~ 0.6, recombinant protein expression was induced overnight at 16 °C using 0.5 mM isopropyl β-D-1-thiogalactopyranoside. Cells were harvested by centrifugation at 3000× *g* for 30 min at 4 °C. The pellet was then suspended in lysis buffer (20 mM HEPES pH 7.4, 500 mM NaCl, Pierce™ Protease Inhibitor Tablets according to the manufacturer’s instructions (1 tablet/10 mL; Thermo Fisher Scientific Inc. Mississauga, ON, Canada)) and the cells were disrupted using an Emulsiflex homogenizer (Avestin Inc, Ottawa, ON, Canada) for 10 min in pulsation mode with pressure set between 10,000 and 15,000 psi. Cell debris was removed by centrifugation at 19,000× *g* for 45 min at 4 °C.

The recombinant protein was further purified from clarified lysates using NiNTA resin. Briefly, cell lysates were passed through NiNTA resin using either a Preppy™ 12-port vacuum system, gravity column, or FPLC system (ÄKTA purifier, Cytiva Life Sciences, Vancouver, BC, Canada) to permit gradient elution. In all cases, the binding buffer was 20 mM HEPES (pH 7.4) and 500 mM NaCl, the elution buffer was the binding buffer plus 500 mM imidazole, and elution was achieved using 5 to 100% elution buffer. The resulting fractions were analyzed by 12% SDS-PAGE; selected fractions were pooled and then exchanged to 20 mM HEPES pH 7.4 using 10kDa Vivaspin 20 column (Vivascience GmbH., Hannover, Germany). Protein concentrations were measured using the Pierce™ BCA Protein Assay Kit (Thermo Fisher Scientific Inc., Mississauga, ON, Canada), and purities were evaluated by SDS-PAGE.

For crystallography, the genes for *Fsp*F52AcXE, *Csp*AcXE, *Pbe*AcXE, *Fre*AcXE, AimAcXE, *Fjo*AcXE and a construct coding for the CBM domain of *Fjo*AcXE (residues 37–186) plus two mutations to introduce to additional methionine residues (Leu62Met and Val113Met) were subcloned into pMCSG53 (coding for N-terminal TEV protease site and His-tag) and expressed in Magic competent *E. coli* BL21(DE3). Cells were grown in ZYP-5052 auto-inducing complex medium [50] by incubating for a few hours at 37 °C followed by transferring to 20 °C for overnight growth, and proteins were purified using Ni-NTA affinity chromatography. Expression of selenomethionine-derivatized proteins was carried out in M9 minimal media following the manufacturer’s instructions (Shanghai Medicilon, Shanghai, China). Expression was induced with isopropyl β-D-1-thiogalactopyranoside (IPTG) at 20 °C when the optical density at 600 nm (OD600) reached 1.2. The overnight cell culture was then collected by centrifugation at 6000× *g*. Cells were resuspended in binding buffer (50 mM HEPES pH 7.5, 500 mM NaCl, 5 mM imidazole, and 5% (*v/v*) glycerol) and lysed with a sonicator, and cell debris was removed by centrifugation at 20,000× *g*. The clarified cell lysate was loaded on a 4 mL Ni-NTA column (Qiagen GmbH, Hilden, Germany) pre-equilibrated with 20 mL of binding buffer. The resin was washed with wash buffer (50 mM HEPES pH 7.5, 500 mM NaCl, 35 mM imidazole, and 5% glycerol (*v/v*)), and the proteins were eluted with elution buffer (50 mM HEPES pH 7.5, 500 mM NaCl, 250 mM imidazole, and 5% (*v/v*) glycerol). The His-tagged proteins were then subjected to overnight TEV cleavage using 50 μg of TEV protease per mg of His-tagged protein and dialyzed overnight against the pre crystallization buffer (10 mM HEPES pH 7.5, 300 mM KCl). The His-tag and TEV were removed by running the protein over the Ni-NTA column. The tag-free proteins were concentrated using a BioMax concentrator (MilliporeSigma, Toronto, ON, Canada) and passed through a 0.2 µm Ultrafree-MC centrifugal filter (MilliporeSigma, Toronto, ON, Canada). The purity of the protein was analyzed by SDS-polyacrylamide gel electrophoresis (Appendix A).

### 3.4. Enzyme Activity Screens

Activity screens using 1 mM *p*NP-linked substrates (*p*NP-α-L-arabinofuranoside, *p*NP-β-D-mannopyranoside, *p*NP-β-D-xylopyranoside, *p*NP-α-D-galactopyranoside, *p*NP-α-L-fucopyranoside, *p*NP-β-D-galactopyranoside, *p*NP-α-D-glucopyranoside, *p*NP-acetate, *p*NP-β-D-glucopyranoside, *p*NP-β-D-cellobioside) and 1 mM 4-MUA were performed at 40 °C in 96-well plates at pH 5.5 (50 mM sodium acetate) and pH 7.0 (50 mM HEPES). The absorbance of reactions containing *p*NP-linked substrates was followed continuously at 410 nm for up to 24 h. After 2 h, reactions containing 1 mM 4-MUA were centrifuged at 3900 rpm and supernatants were transferred to a new 96-well plate prior to measuring absorbance at 354 nm. The corresponding standard curve was prepared using eight points between 0.5 and 2.5 mM of 4-MU.

Glycoside hydrolase activity was evaluated using 0.6 mg/mL (total reaction volume of 200 µL) of 16 different polysaccharides (Appendix A), where reactions were performed for up to 24 h at 40 °C in 96-well plates and three pH values: pH 5.5 (50 mM sodium acetate), pH 7.0 (50 mM HEPES), and pH 8.5 (50 mM HEPES). Following incubation, glycoside hydrolase activity was measured using the PAHBAH assay [51]. Briefly, 10 µL of each reaction were transferred to 200 µL of PAHBAH reagent (4-hydrobenzoic acid hydrazide) and samples were incubated at 40 °C for 30 min with shaking prior to measuring absorbance at 405 nm.

For all screens, reactions were performed in triplicate and contained 10% *w/w* of enzyme to dry weight of substrate.

### 3.5. Optimum Reaction Conditions

The pH optimum of each purified protein was tested using 50 mM Universal buffer (67 mM NaCH_3_COO, 67 mM H_3_BO_3_, 67 mM H_3_PO_4_, adjusted to pH values between 3.5 and 9.0 using NaOH) and 0.5 mM 4-MUA. Reactions (200 µL) were initiated by adding 0.02 µg of enzyme (0.03 µg for *Fsp*AcXE) and continued for 30 min at 40 °C. 4-MUA was selected to determine the pH optimum of each enzyme given the relative stability of this substrate under alkaline conditions [29]. For all reactions using 4-MUA, absorbance was measured at 354 nm and the reaction mixture without enzyme was used as a blank.

The temperature stability of each enzyme was tested against 0.5 mM 4-MUA in 50 mM Universal buffer adjusted to the respected optimum pH. The reaction was started by addition of 0.02 µg of protein (0.03 µg for *Fsp*AcXE), and each suspension was incubated for up to 24 h at 20, 40, 55 and 70 °C. The total reaction volume was 200 µL.

### 3.6. Activity on Acetylated Xylooligosaccharides

The activities of studied AcXEs on acetylated xylooligosaccharides were tested using highly branched acetylated and feruloylated CF-XOS (described in Appeldoorn et al., 2013 [30]) and a per-acetylated xylo-oligosaccharide mixture (DP 4–7). Acetic acid release was determined using the acetic acid kit (K-ACETRM, Megazyme). Reactions (50 µL final volume) containing CF-XOS comprised 1 µg enzyme, and 50 mM HEPES buffer (pH 7.0) and 0.5 µg of CF-XOS, and were incubated at 30 °C for 20 min and 4 h. Reactions (60 µL final volume) containing the per-acetylated xylo-oligosaccharide mixture comprised 2 µg enzyme, 50 mM HEPES buffer (pH 7.0) and 0.5 µg of per-acetylated xylo-oligosaccharide mixture, and were incubated at 30 °C for 20 min, 4 h and 20 h. The maximal acetic acid release from each substrate was determined using a modified method described by Teleman et al., 2002 [4]. Briefly, 1% (*w/v*) substrate was suspended in 50 µL 0.5 M NaOH for 30 min at 70 °C and 450 rpm. The reaction solutions were then neutralized, and the release of acetic acid was measured with acetic acid kit. Control reactions without enzyme were run in parallel and the amount of non-enzymatically released acetic acid was subtracted from the values of enzyme–substrate reaction mixtures.

### 3.7. Complementation Assay Using AcXE and α-Glucuronidase

Activity levels of the studied AcXE and *Axy*Agu115A (GH115) on acetylated glucuronoxylan were tested alone and in pairs. In total, 1% (*w/v*) substrate and 0.5 µg of enzyme were incubated for 20 min in 50 mM HEPES pH 7.0 at 30 °C. The total reaction volume was 25 µL. Acetic acid release was detected using Acetic acid kit and release of glucuronic acid using D-glucuronic/D-galacturonic acid kit (Megazyme).

### 3.8. Protein Crystallization, Data Collection, and Structure Determination

All crystals were grown at room temperature using the vapor diffusion sitting drop method. For the selenomethione-derivatized native *Csp*AcXE crystal, 20 mg/mL protein was mixed with 20 mM acetylated xylooligosaccharides, and then with reservoir solution 30% (*w/v*) Jeffamine ED-2001 pH 7.0. For the selenomethione-derivatized *Pbe*AcXE·MeGlc*p*A-Xyl*p* complex crystal, 20 mg/mL protein was mixed with 20 mM acetylated xylooligosaccharides, and then with reservoir solution 0.1 M Tris pH 6.5 and 20% (*w/v*) PEG 3350; for the selenomethione-derivatized *Pbe*AcXE·acetate complex crystal, 20 mg/mL protein was mixed with 70 mM xylose, then with reservoir solution 0.1 M Tris pH 8.5, 0.2 M sodium acetate, and 30% (*w/v*) PEG4K. For the native *Pbe*AcXE crystal, 20 mg/mL protein was mixed with 20 mM xylobiose, and then reservoir solution 0.2 M magnesium chloride and 25% (*w/v*) PEG3350. For the selenomethione-derivatized *Fjo*AcXE CBM Leu62Met + Val113Met mutant, 15 mg/mL protein was mixed with reservoir solution 1 M ammonium sulfate, 0.6 M sodium citrate and 0.1 M calcium chloride. All crystals were cryoprotected with paratone oil.

Diffraction data at 100 K were collected at beamline 21-ID of the Life Sciences Collaborative Access Team at the Advanced Photon Source, Argonne National Laboratory. Diffraction data were processed using HKL3000 [52]. Structures were solved by single anomalous dispersion (SAD) phasing using Phenix.AutoSol [53]. All model building and refinement were performed using Phenix.refine and Coot [54] with full B-factor anisotropy for all structures but the *Fjo*AcXE CBM, which was refined with TLS parameterization. Atomic coordinates have been deposited in the Protein Data Bank with accession codes 7TOH, 7TOI, 7TOJ and 7TOK.

## 4. Conclusions

Six *Fjo*AcXE-like enzymes were biochemically characterized, and solved structures were obtained for two of these (*Pbe*AcXE and *Csp*AcXE), revealing structure-functional relationships for AcXEs with potential to fully deacetylate glucuronoxylans. *Fjo*AcXE-like AcXEs comprise a novel CBM-like domain and an SGNH domain. CBM-like domains share structural similarity to several types of β-structures, including CBM1 and CBM35. However, the CBM-like domains of *Fjo*AcXE-like AcXEs do not clearly group with existing CBM families. The complex structure of *Pbe*AcXE with bound MeGlc*p*A-Xyl*p* reveals interactions between the enzyme and xylan side groups and the elements that line the substrate specificity. The substrate is bound at the positively charged interface between the SGNH and CBM-like domains consisting of conserved amino acid residues. The MeGlc*p*A is bound by Arg139 in a bidentate way, and we suggest that fully conserved Arg139 is crucial for the substrate specificity of *Fjo*AcXE-like enzymes. Other amino acid residues that bind to hydroxyl groups of MeGlc*p*A are Arg25 and Asn26. The Xyl*p* unit is bound to Ser188, Arg193 and His366. Two distinguishing features of the active site include the catalytic serine in *Fjo*AcXE and its homologs that occur in a novel GNSI sequence motif, and the absence of the GXSXG consensus sequence of α/β hydrolases. Together, the biochemical and structural characterization of *Fjo*AcXE and its homologs open the way to establishing a new carbohydrate esterase family.

## Figures and Tables

**Figure 1 molecules-27-02655-f001:**
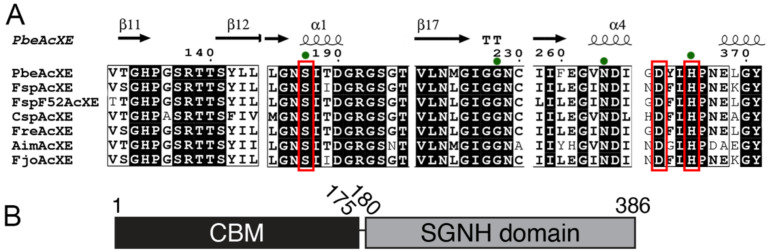
(**A**) Partial alignment of selected sequences and the biochemically characterized *Fjo*AcXE from *Flavobacterium johnsoniae* (Razeq et al., 2018) [27]. Catalytic residues are rounded with red and the SGNH residues of the SGNH blocks are marked with green dots above columns. The amino acid numbering showed above columns are from *Fsp*AcXE sequence, including signal sequence. (**B**) A graphical presentation of the domain architecture of *Pbe*AcXE.

**Figure 2 molecules-27-02655-f002:**
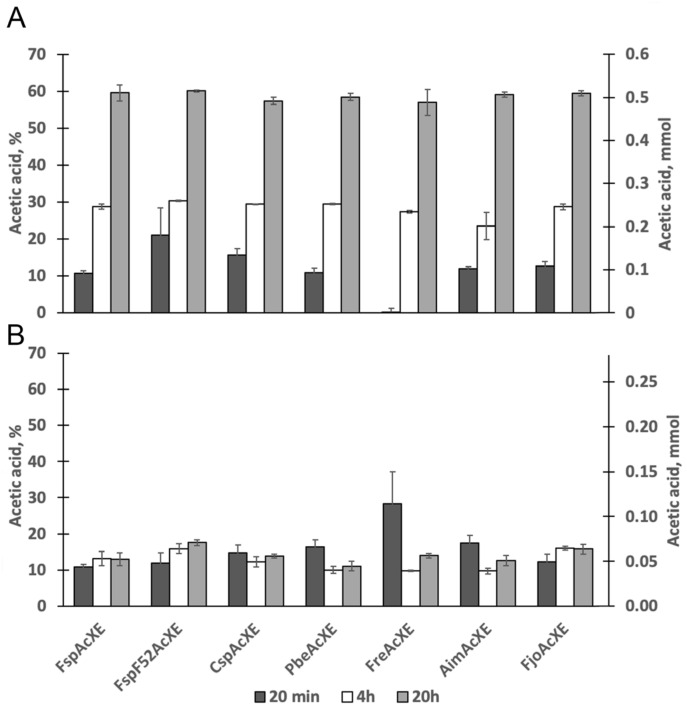
Acetic acid release from (**A**) per-acetylated xylooligosaccharide (DP 4–7) using 2 μg of enzyme, and (**B**) from CF-XOS using 1 μg enzyme. Reactions (60 μL in Subfigure (**A**) and 50 μL in Subfigure (**B**)) were performed at 50 mM HEPES and pH 7.0, and were incubated for 20 min, 4 h, and 20 h. Acetic acid release was detected with acetic acid kit (K-ACETRM, Megazyme). Error bars represent standard deviation (*n* = 3).

**Figure 3 molecules-27-02655-f003:**
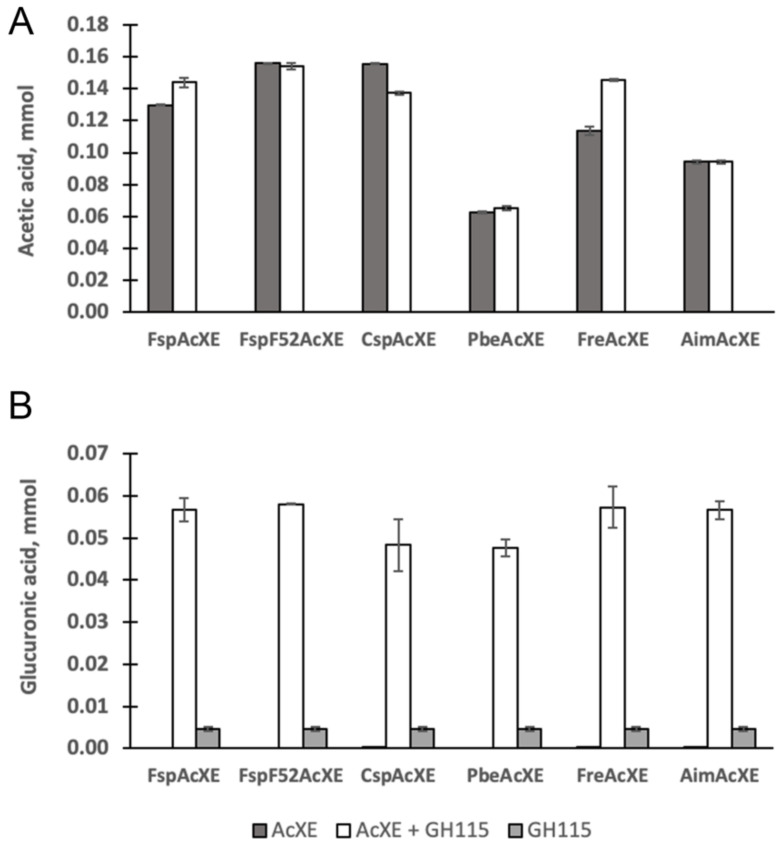
Release of acetic acid (**A**) and glucuronic acid (**B**) from acetylated Eucalyptus wood (AcXOS). *n* = 3; error bars correspond to standard deviation.

**Figure 4 molecules-27-02655-f004:**
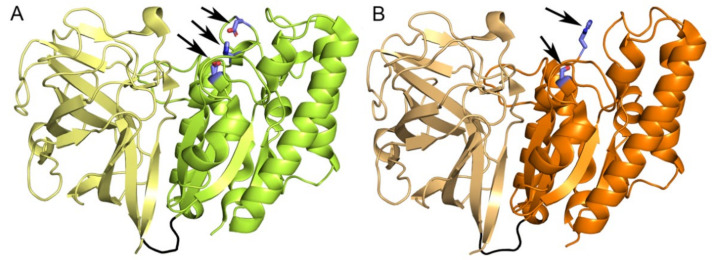
(**A**) A Crystal structure of acetyl xylan esterase from *Prolixibacter bellariivorans*. The N-terminal CBM domain is shown in pale yellow, SGNH domain in light green, and linker between them in black. The amino acid residues of the catalytic triad (Ser188, Asp363, and His366) are shown in blue sticks. (**B**) The crystal structure of acetyl xylan esterase from *Chryseobacterium* sp. The N-terminal CBM domain is shown in light brown, SGNH domain in orange, and linker in black. Catalytic Ser189 and Asp363 are shown in sticks.

**Figure 5 molecules-27-02655-f005:**
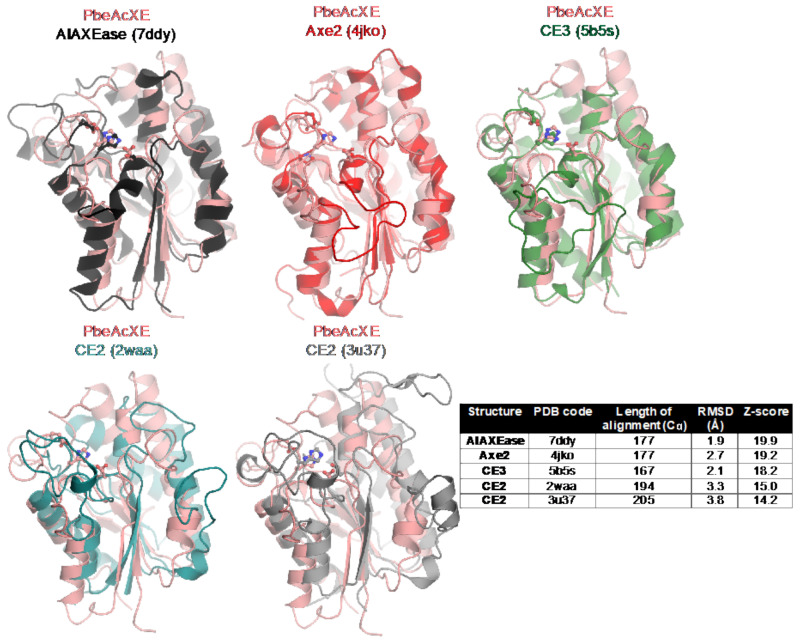
A crystal structures of acetyl xylan esterase from *Prolixibacter bellariivorans* (*Pbe*AcXE) is superimposed with *Al*AXEase from *Arcticibacterium luteifluviistationis* (PDB ID 7ddy), Axe2 from *Geobacillus stearothermophilus* (PDB ID 4jko) [33], a CE3 enzyme from *Talaromyces cellulolyticus* (PDB ID 5b5s) [34], and CE2 enzymes from *Cellvibrio japonicus* (PDB ID 2waa) [35] and *Butyrivibrio proteoclasticus* B316 (PDB ID 3u37) [36]. The table shows the structurally most similar structures found through a Dali search.

**Figure 6 molecules-27-02655-f006:**
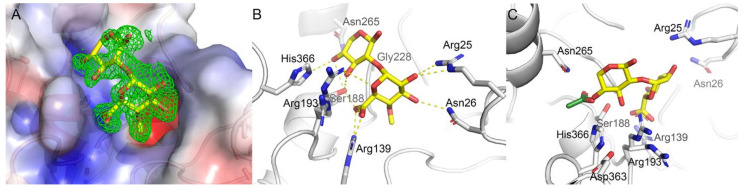
The complex crystal structures of acetylxylan esterase from *Prolixibacter bellariivorans* (*Pbe*AcXE) with methylated D-glucuronic acid linked to D-xylopyranose (MeGlc*p*A-(1→2)-Xyl*p*). (**A**) The polder omit electron density map for the ligand is shown at 3σ contour level in green. Surface charges are shown in blue (positive charge) and red (negative charge). (**B**) The amino acid residues at the active site that participate in ligand binding in *Pbe*AcXE are labelled and shown in sticks. The hydrogen bonds between these residues and the ligand are marked as yellow dash lines. (**C**) Superposed complex structures of *Pbe*AcXE with MeGlc*p*A-(1→2)-Xyl*p* and acetate. Acetate is shown in green. The catalytic Ser188 exists in the same conformation in (**B**,**C**).

**Figure 7 molecules-27-02655-f007:**
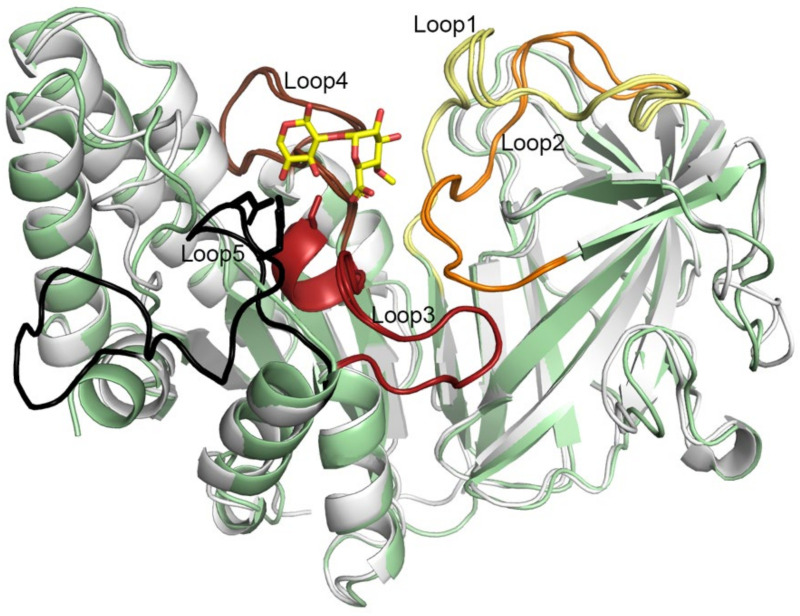
Overlaid crystal structures of acetyl xylan esterases from *Prolixibacter bellariivorans*, (gray) in complex with MeGlc*p*A-(1→2)-Xyl*p* ligand (yellow) and *Chryseobacterium* sp. YR480 (light green). The five conserved loops in the AcXEs are shown in yellow, orange, red, brown and black. The catalytic serine is shown in sticks.

**Table 1 molecules-27-02655-t001:** Percentage of hydrolyzed substrate by the studied acetylxylan esterases in 2 h using 1 mM substrate and 40 °C temperature (*n* = 3).

Substrate	pH	*Fsp*AcXE	*Fsp*F52AcXE	*Csp*AcXE	*Pbe*AcXE	*Fre*AcXE	*Aim*AcXE
*p*NPA	5.5	23	27	52	42	47	24
	7	47	47	57	55	50	53
4-MUA	5.5	35	32	33	31	25	25
	7	45	42	48	47	40	54

## Data Availability

Structure factors and atomic coordinates of the solved crystal structures are deposited in the Protein Data Bank (PDB) with accession IDs 7TOG, 7TOH, 7TOI, 7TOJ, and 7TOK, from which the data might be downloaded.

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
