# Peer review of "Elucidating Sequence and Structural Determinants of Carbohydrate Esterases for Complete Deacetylation of Substituted Xylans"

_molecules, 2022, doi:10.3390/molecules27092655_

Round 1

Reviewer 1 Report

The paper ‘Elucidating sequence and structural determinants of carbohydrate esterases for complete deacetylation of substituted xylans’ by Leena Penttinen et al. deals with the production of six FjoAcXE homologues, their biochemical characterization and the crystal structure determination of some of them.

The study aims at revealing structure-functional relationships for AcXEs with potential to fully deacetylate glucuronoxylan, making the hypothesis, based on sequence and structure comparative analysis, that the sequences of FjoAcXE and its homologues might comprise a new carbohydrate esterase family.

Furthermore, the comprehension of the catalytic mechanisms of this class of enzymes that selectively remove xylan side groups can broadly impact xylan applications, from use as packaging films and nutrient additives, to fuels and chemicals.

The cloning/purification effort of six proteins is indeed to be acknowledged for the evaluation of this paper.

The readability of the paper is overall sufficient. It has a proper length and is reasonably well structured.

Crystal structures are critically and systematically well analyzed/compared.

Literature appears to be adequately cited and information needed to reproduce all experiments are well detailed.

Overall, the work is sound, and the experimental results support the conclusions drawn.

Still, I have one concern. I think authors have quite neglected to frame the work by devoting more space to the description of practical potential applications of these enzymes. Just a couple of lines are devoted to this at the beginning. I find that the systematic description of the homologues was of course due, but the paper would benefit from the addition of a clearer applicative scope for this research; this would also make it attractive to a more general audience.

Then some questions:

Was FspAcXE crystallization attempted? If not, why?

Have the authors an explanation for residues 350 to 365 in CspAcXE having unclear or missing electron density? Their position might be critical since they flank the active site.

Are the residues that have no density in the crystallized construct of FjoAcXE critical for its function?

From the crystallographic point of view, I have some concerns to point out:

Why was the parametrization used in refinement TLS for all structures? Four out of five structures have near atomic resolution. I would suggest to use full B-factor anisotropy for these four and TLS for the low resolution one.

RMSD on bond length is almost one order of magnitude higher in the near resolution structures rather than in the low resolution one. These values are still acceptable…but have authors got a possible explanation for this?

Figure 4A and 4B have residues that are indicated with arrows. I don’t know if it is a problem with the pdf I am reviewing but I think those residues are almost not visible. I think the whole color coding of the figure should be changes and, also the thickness of the sticks of the highlighted residues. Maybe the figure would also be improved if the two structures were represented in stereo.

Concerning Figure 5, the ribbons toward the backplane are barely visible. Furthermore, for the sake of a more formal comparison, it would be good to have a chart of the backbone atoms superposition vs residues for all the structures (maybe in the supplementary material).

Minor points/Typos:

I find more appropriate the term ‘Multiplicity’ rather than ‘Redundancy’ Table S5.

Page 5, line 150: belog should read belong

Author Response

Dr. Jurak

Reviewer 2 Report

The manuscript is another important contribution to current knowledge of enzymes involved in plant cell wall degradation. The manuscript presents for the first time convincing data that the acetylxylan esterase liberating 3-O-acetyl group on MeGlcA-substituted xylopyranosyl residues in plant xylans (discovered recently by these researchers), occurs widely and that one of the substrate specificity factors is the double ionic interaction of the MeGlcA carboxyl group with an arginine that is conserved in the enzymes. This fact is documented also by the first 3D structures of these esterases with aldouronic acid ligand. The results presented in the reviewed manuscript open the way to establishment of a new carbohydrate esterase family. Maybe this could be mentioned in the Conclusion.

My major criticism concerns the experiments with per-O-acetylated xylooligosaccharides of DP 4-7. Are these oligosaccharides soluble in water? How was the substrate solution containing 0.5 micro-g per 60 micro-L of the reaction mixture prepared? Calculation based on this data shows that concentration of the acetylated xylooligosaccharides was 8.33 micro-g/mL while enzyme concentration was 2% (w/v), which means 20 mg/ml. More surprising is the information on p. 3, line 112, that maximal yield of acetic acid could be 3 mg/ml. It is obviously a miscalculation (maybe a correct value is 3 micro-g/mL). The most serious problem of the sections 2.3. and 3.6. is a lack of control values determined in mixtures in the absence of the enzymes. The authors should at least mention that controls were run in parallel and either no release of acetic acid was observed or the amount released spontaneously in the control mixtures was subtracted. Taking the extraordinarily high and non-physiological concentration of the enzyme as well as its ratio to the substrate together with  the questions concerning the substrate solubility and possibility of the non-enzymatic release of the acetic acid at pH 7, the manuscript would gain if the section with per-O-acetylated xylooligosaccharides would be completely eliminated. The information is not needed since the data on cooperativity of the new AcXEs with apha-glucuronidase presented  in Fig. 3 is much more important.

The same miscalculation applies to the content of acetic acid in corn fiber oligosaccharide fraction. The substrate concentration was 0.5 micro-g per 50 micro-L, i.e. 10 micro-g/mL, and the enzyme concentration 1% (10 mg/mL), i.e. non-physiological. According to the characterization of oligosaccharides released from corn fiber (Ref. 48), weight percentage of acetic acid should not exceed 5 or 10%. Therefore, maximum releasable amount of acetic acid should not exceed 1 micro-g/mL. Moreover, the authors should take in consideration that the acetic released by the enzymes may not originate only from position 2 vicinal to 3-linked arabinose since the enzyme efficiently works as 3-deacetylase (Refs. 25-26).

  1. 259 –catalytic Ser188 is bound to O-3 (not to endocyclic O-5) atom of Xylp residue (Fig. 7B). It is the oxygen atom, from which acetyl group is deesterified. Concerning the enzyme structures, for review purposes I would appreciate sending me or making accessible the PDB-files of the structures deposited in RCSB protein database. In the PbeAcXE complex with acetate (Fig. 7C), its location is close to position 4 (l. 275) of the superimposed Xylp residue. I agree that this would be consistent with the recognition of 4-acetyl group at the non-reducing end Xylp residue 2-substituted with MeGlcA (l. 276). However, to my knowledge such an activity has never been tested or demonstrated for this type of acetylxylan esterases. Nevertheless, the authors mention that the catalytic Ser188 was observed in two conformations. It could be indicated whether the conformation of Ser188 shown in Fig. 7B is the same as that in Fig. 7C. In any case, it is another reason to inspect the PDB-files and estimate the consequences of various conformations of the Ser188 in terms of catalytic versatility of PbeAcXE.
  2. 137 – the authors should be more careful about the statement that 10-fold increase in the released glucuronic acid was due to the deacetylation of vicinal 3-hydroxy group. The AcXEs studied presumably efficiently catalyze such a deesterification (based on the properties of FjoAcXE, Ref. 25), therefore, such an explanation is most plausible. However, there is also a possibility that the enhancement of deglucuronidation is due to a removal of 3-acetyl group from neighbouring xylopyranosyl residue

In Fig. S3 as well as Fig. 1 I would appreciate showing secondary structure elements. It would facilitate reading of the maintext, in particularly sections 2.6 to 2.8, of which the section 2.6 could be typed as the last one. For clarity reasons, Fig. 1 could be combined with a graphical presentation of subdomain structure, showing N-terminal CBM-like subdomain (amino acids 1-175) linked to  C-terminal SGNH subdomain (amino acids 180-385) (PbeAcXE numbering)

Formal errors:

It is recommended to use an abbreviation for the used fraction of esterified corn fiber oligosaccharides, e.g. CFO or CFXO, since the text contains various expressions for this material, including just incorrect “corn fiber” (p. 12, lines 401, 402).

Two-three letter abbreviations of the names of microorganisms serving as a source of the enzymes should be in italics (throughout the manuscript). Example: FjoAcXE should appear as FjoAcXE.

The legend to Table 1 would sound better if the authors use simply just “Percentage of the substrate hydrolyzed…” instead of “Relative release (%) of acetic acid…”. The values given in the table are based on enzymatic determination of released acetic acid, but “Relative” to what?

Spaces are frequently missing before insertion of references in square brackets.

  1. 3, line 102 – insert a symbol for the temperature degree.
  2. 136, 411 – presumable alpha-glucuronidase, not glucuronoyl esterase

Fig. S1 – how the authors explain a lower molecular weight of PbeAcXE (lane 4)?

Please, expand the legends to tables S3 and S4 in the supplementary material, so they would become more self-explanatory. By the way, the data are not convincing (particularly in Table S4) since they are quite scattered. They show just the trends. Moreover, the data in Table S3 does not correlate satisfactorily to those in Table 1.

In Table S2 indicate a supplier of corn fiber oligosaccharide fraction. This material could also be, at least partially, characterized or described. The name “fraction B 28” (line 301) does not say anything, even with a reference to Appeldoorn et al., 2013. Eucalyptus acetylglucuronoxylan should also be included in the table, if it is kept in supplementary material. Moreover, in my opinion all the substrates should be listed in the maintext. In contrast, acetylgalactoglucomannan and its supplier are indicated in the maintext, although no results are reported.

Table S7 is useless. It shows just a single positive result (with 4-nitrophenyl acetate) that is presented in the maintext (Table 1). The negative results could be summarized in a single sentence that the enzyme did not exhibited an activity on any nitrophenyl glycosides tested (and listed in the experimental section)

Conclusion: minor revision

Author Response

Dr. Jurak

Round 2

Reviewer 1 Report

The points raised by this reviewer have been satisfactorily addressed and the paper containing the modifications proposed can now be published.